# Unifying Graph-Based and Pairwise-Based Representations for Gene Regulatory Network Inference from scRNA-seq Data

## Abstract

Gene regulatory networks (GRNs) capture the underlying interactions through which transcription factors (TFs) regulate genes. Based on gene expression data, existing GRN inference approaches generally fall into two categories: graph-based methods, which model the GRN as a whole graph, and pairwise-based methods, which decompose the GRN into individual TF–target gene pairs for modeling. However, each approach exhibits limitations that are precisely the strengths of its counterpart. Graph-based methods tend to overfit due to their reliance on a single training graph, compared to the numerous TF–target gene pairs available for pairwise-based methods during training. In contrast, pairwise-based methods overlook the global topological structure, which is essential to graph-based learning. To address these limitations, we propose scUniGP, a unified framework that jointly models global regulatory topology and local TF–target interactions. scUniGP first extracts multi-scale topological features from the whole regulatory graph, and then hierarchically integrates these global representations with local features derived from pairwise modeling for comprehensive GRN inference. Extensive experiments on seven benchmark datasets demonstrate that our model consistently achieves state-of-the-art performance, validating the effectiveness of our integrative design.

## 1 Introduction

Gene regulatory networks (GRNs) describe the intricate relationships by which transcription factors (TFs) control the expression of their target genes Marbach et al. (2010). A GRN is represented as a directed graph, where nodes denote TFs or genes and edges indicate regulatory relationships. As shown in Figure 1(a), for the transcription factor $g_1$ and gene $g_5$, the directed edge $g_1 \rightarrow g_5$ implies that higher expression of $g_1$ leads to increased expression of $g_5$, reflecting the regulatory mechanism between them Hou et al. (2020); Zhang et al. (2023). Thanks to the development of single-cell RNA sequencing (scRNA-seq), which provides accurate expression profiles of genes and TFs across individual cells, researchers can now investigate GRNs at the cell level by using single-cell expression data Reuter et al. (2015); Tanay & Regev (2017); Xu et al. (2023).

However, current scRNA-seq datasets pose two major challenges for GRN inference. First, since regulatory annotations require specialized biological knowledge, existing datasets often suffer from incomplete edge annotations Badia-i Mompel et al. (2023). Second, scRNA-seq typically measures tens of thousands of genes, more than the number of cells, resulting in an extremely large GRN with limited training samples Risso et al. (2018); Sun et al. (2025). Therefore, recent methods have moved beyond traditional statistical approaches and instead adopt deep learning methods specifically designed for GRN inference. These methods can be categorized into two main groups: graph-based methods and pairwise-based methods.

Graph-based methods, as shown in Figure 1(b), treat the GRN as a whole graph, leveraging graph neural networks (GNNs) to capture global topological relationships among transcription factors and genes Chen & Liu (2022b); Guo et al. (2023); Bai & Wang (2024). Their principal strength lies in mitigating the challenges of incomplete annotations through the message-passing mechanisms inherent to GNNs Kipf & Welling (2017). For example, let $g_1$ and $g_2$ denote two different TFs,

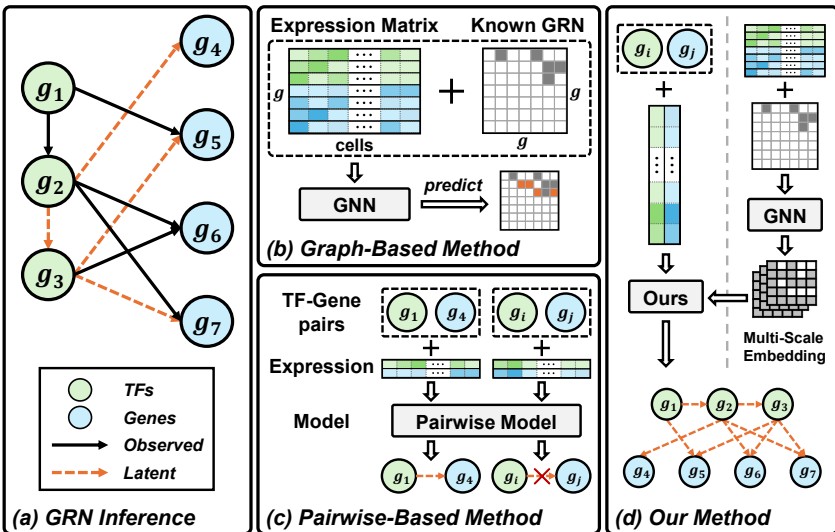

Figure 1: Illustrations of GRN inference and related approaches. (a) GRN inference aims to leverage observed regulatory relationships (black arrow) to predict unknown ones (orange dashed arrow). (b) Graph-based methods model the entire GRN as a graph, using observed edges (gray blocks) to predict unknown edges (orange blocks) in the adjacency matrix. (c) Pairwise-based methods decompose the GRN into individual TF–gene pairs and predict their relationships independently. (d) Our method integrates both paradigms to enable comprehensive GRN inference.

and let $g_3$ denote a target gene. If $g_1 \rightarrow g_3$ and $g_2$ share similar global topological features with $g_1$, graph-based methods can potentially learn the regulation $g_2 \rightarrow g_3$ even without explicit annotation. However, graph-based methods typically trained on a single GRN graph, are prone to overfitting Wu et al. (2021). Moreover, GNNs may overlook important local interactions with the vast number of genes (nodes) during global information propagation Grønbech et al. (2020). These limitations hinder the performance of graph-based models in practice.

Pairwise-based methods, as shown in Figure 1(c), decompose the entire GRN into individual TF-target gene pairs, and learn each regulatory interaction independently KC et al. (2019); Wang et al. (2024). By generating a significantly larger training set, these methods mitigate the overfitting problems inherent to graph-based methods. Moreover, they effectively capture fine-grained regulatory interactions and exhibit strong predictive performance in practice Cho et al. (2016); Greener et al. (2022). However, because pairwise-based methods ignore the global network structure, they suffer from incomplete annotations, and they also fail to capture higher-order dependencies Wang et al. (2021). For example, if transcription factor $g_1 \rightarrow g_2$, and $g_2 \rightarrow g_3$, the indirect regulatory influence of $g_1$ on $g_3$ can be overlooked by pairwise-based methods. These limitations hinder comprehensive GRN inference over the full regulatory network and constrain the model's generalizability.

To leverage the strengths and mitigate the weaknesses of both methods, as shown in Figure 1(d), we propose *scUniGP* (Unified Graph-Pairwise model), a novel framework that integrates topological knowledge from regulatory graphs with Transformer-based representations of TF-target gene pairs. *scUniGP* employs a two-stage architecture. In the first stage, a graph neural network processes gene expression matrix and prior regulatory networks to learn multi-scale gene embeddings, capturing hierarchical global features and producing edge-level confidence scores as expert guidance. In the second stage, a Transformer encodes expression profiles of candidate TF-target pairs and fuses them with previously learned GNN embeddings at multiple levels, ensuring that fine-grained expression patterns are progressively aligned with the broader regulatory topology context. This hierarchical integration not only overcomes the overfitting issues of graph-based approaches and the locality constraints of pairwise-based methods, but also produces unified representations that reflect both global context and local interaction, thereby facilitating accurate and comprehensive GRN inference. We evaluate *scUniGP* on seven widely used benchmark datasets in comparison with state-of-the-art graph-based and pairwise-based methods. Experimental results show that *scUniGP* consistently

achieves superior performance across all datasets, validating the effectiveness of our integrative design and the importance of combining global and local modeling in GRN inference.

Our main contributions are summarized as follows:

- We performed a deep analysis of the key challenges in GRN inference from scRNA-seq data, categorized existing approaches into graph-based and pairwise-based methods, and identified their respective strengths and limitations, revealing their complementary nature.
- Based on these insights, we propose *scUniGP*, a novel framework that integrates global topology modeling with local interaction learning to leverage their complementary advantages and address their shortcomings.
- Extensive experiments on seven widely used GRN benchmark datasets demonstrate that *scUniGP* consistently outperforms state-of-the-art methods, validating the rationale behind our findings and the effectiveness of our integrative design.

## 2 RELATED WORK

**Traditional Methods.** Early gene regulatory network (GRN) inference methods can be broadly categorized into three classes: information-theoretic models, differential equation-based frameworks, and Boolean networks. Correlation-based approaches, such as LEAP Specht & Li (2017) and mutual information methods Song et al. (2012), assume that transcription factors and their targets exhibit coordinated expression patterns across conditions. Differential equation-based models, such as SCODE Matsumoto et al. (2017), utilize pseudotime to capture temporal dynamics via ordinary differential equations. Boolean networks Dorier et al. (2016); Schwab et al. (2020) abstract gene states into binary logic, offering interpretable models validated through dynamic simulations. With the rise of scRNA-seq, statistical learning techniques such as regression and probabilistic graphical models have gained traction Cho et al. (2016). SINCERITIES Papili Gao et al. (2018) employs ridge regression to model time-dependent changes in gene expression. Inspired by GENIE3 Huynh-Thu et al. (2010), which models GRN inference via random forest-based regression, GRNBoost2 Moerman et al. (2019) and DIRECT-NET Zhang et al. (2022) adopt gradient boosting trees to improve computational efficiency and regulatory link reliability. Bayesian models like scMTNI Zhang et al. (2023) further integrate prior knowledge, though their reliance on Gaussian expression assumptions may limit applicability. To address the scalability, noise, and nonlinearity challenges of scRNA-seq, deep learning has been applied to GRN inference, mainly through **graph-based methods** that exploit network structure and **pairwise-based methods** that model TF–gene interactions.

**Graph-based Methods.** Given the intrinsic graph structure of gene regulatory networks, many approaches have employed graph neural networks (GNNs) to learn regulatory dependencies. DeepSEM Shu et al. (2021), inspired by structural equation modeling Yu et al. (2019), encodes GRNs as latent adjacency matrices within a variational autoencoder framework. Chen *et al.* proposed GENELink Chen & Liu (2022b), which uses graph attention networks (GATs) to learn gene embeddings by integrating expression data and prior topology. GNNLink Guo et al. (2023) extends this idea with graph convolutional networks (GCNs) and matrix completion to mitigate dropout effects, while GRNNLink Bai & Wang (2024) replaces GCNs with graph recurrent neural networks to enhance robustness and capture dynamic dependencies. These methods leverage global topological signals, but are often limited by sparse edge annotations and single-graph supervision.

**Pairwise-based Methods.** To alleviate the limitation of sparse training samples, some methods treat GRN inference as a pairwise classification task. GNE KC et al. (2019) combines prior networks and gene expression using multi-layer perceptrons (MLPs) to embed gene pairs. CNNC Yuan & Bar-Joseph (2019) transforms co-expression histograms into image-like representations for CNN-based classification, though at significant computational cost. More recently, scGREAT Wang et al. (2024) introduces a Transformer-based framework inspired by NLP, learning TF–gene pair embeddings in a context-aware manner from gene expression and biological annotations. While pairwise methods scale well and generate abundant training samples, they often lack global structural awareness and struggle to capture higher-order regulatory patterns. Despite their respective strengths, graph-based and pairwise-based methods suffer from complementary limitations in capturing multi-scale regulatory dependencies. Motivated by this, we propose *scUniGP*, a unified framework that integrates global topology modeling with local interaction learning for robust GRN inference.

## 3 METHOD

### 3.1 PRELIMINARIES

**GRN inference**. We represent the gene regulatory network as the directed graph $(G, E)$, where $G = \{g_i\}_{i=1}^n$ denotes the set of transcription factors and target genes (i.e., the nodes), and $E = \{(g_i, g_j)\} \subseteq G \times G$ denotes the set of regulatory interactions (i.e., the edges), where $g_i \to g_j$ denotes that $g_i$ regulates $g_j$. Note that we employ a slight abuse of notation, using $g$ to refer to both TFs and genes, without loss of generality Guo et al. (2023). The goal of GRN inference is to recover the unknown edges.

Given single-cell RNA sequencing data from $c$ cells, every cell contains expression values corresponding to genes in $G$. For example, for cell $i$, its gene expression values are represented as $\boldsymbol{v}_i = (v_{i,1}, v_{i,2}, \ldots, v_{i,n})$, $n = |G|$. Then, the RNA-seq data could be represented as the expression matrix $\boldsymbol{X} = [\, v_{i,j} \,]_{i=1,\ldots,c}^{j=1,\ldots,n} \in \mathbb{R}^{c \times n}$. In practice, define $y_{ij} \in \{0, 1\}$ as the ground-truth label for the edge $(g_i, g_j)$ to indicate whether $g_i$ regulates $g_j$, and split $E$ into training set $E_{tr}$ and test set $E_{te}$, the objective for GRN inference is to minimize:

$$\mathcal{L}_{\mathrm{GRN}} = \sum_{(g_i, g_j) \in E_{\mathrm{tr}}} \mathcal{L}\left(m\left(\boldsymbol{X}; E_{tr}\right), y_{ij}\right) \tag{1}$$

where $m$ is the trainable model to extract features and predict $y_{ij}$ and $\mathcal{L}$ denotes the loss function.

**Graph-based methods** regard the entire GRN as a single graph to train a Graph Neural Network (GNN). Each node embedding is directly obtained from the expression matrix: $e_k = \boldsymbol{X}_{:,k} \in \mathbb{R}^c$ for gene $g_k$. Then, GNN propagates node information along the training edges $E_{\mathrm{tr}}$, yielding predictions for each candidate edge. The training loss can be written as:

$$\mathcal{L}_{\mathrm{Graph}} = \sum_{(g_i, g_j) \in E_{\mathrm{tr}}} \mathcal{L}_{\mathrm{BCE}}\left(m_{\mathrm{GNN}}(\{e_k\}_{k=1}^n; E_{\mathrm{tr}})_{ij}, y_{ij}\right), \tag{2}$$

where $m_{\mathrm{GNN}}$ denotes the graph neural network, $m_{\mathrm{GNN}}(\cdot)_{ij}$ represents the $(i, j)$-th entry in the predicted adjacency matrix of the graph, and $\mathcal{L}_{\mathrm{BCE}}$ is the binary cross-entropy loss.

As shown in Eq. equation 2, the loss calculation involves all nodes and edges, and only a single graph is available to train the GNN, introducing the aforementioned overfitting problem and thereby hindering the performance.

**Pairwise-based methods** decompose GRN into independent TF–target gene pairs for the training. For each pair $(g_i, g_j)$, they first get their expression embeddings from $\boldsymbol{X}$, e.g., $(e_i = \boldsymbol{X}_{:,i}, e_j = \boldsymbol{X}_{:,j})$ and encode them via a shared encoder. Then, the embeddings are fused to predict the corresponding label $y_{ij}$. The training loss could be written as:

$$\mathcal{L}_{\mathrm{Pairwise}} = \sum_{(g_i, g_j) \in E_{\mathrm{tr}}} \mathcal{L}_{\mathrm{BCE}}\left(m(e_i, e_j), y_{ij}\right), \tag{3}$$

where $m$ denotes the model for encoding gene embeddings.

By generating a much larger set of training samples, pairwise-based methods mitigate the overfitting problem. However, as shown in Eq. equation 3, they totally ignore the network structure $E_{\mathrm{tr}}$ and fail to model global information, which limits their generalizability.

### 3.2 OUR MODEL—SCUNIGP

**Overview**. To overcome the limitations of graph-based methods and pairwise-based methods, we propose *scUniGP*, a unified hybrid framework that seamlessly blends global regulatory topology with fine-grained expression dynamics. There are three interconnected components: (1) Global Branch, which derives multi-scale gene embeddings and edge-level confidence scores from a prior regulatory graph. (2) Pairwise Branch, which applies a Transformer encoder to the expression profiles of each candidate TF–target pair. (3) Multi-scale Fusion, which progressively integrates the GNN-derived features into the pairwise outputs to yield a consolidated prediction.

**Global Branch**. Given the training gene regulatory graph $(G, E_{tr})$, $|G| = n$, the adjacency matrix $\mathbf{A}_{tr} \in \{0, 1\}^{n \times n}$ and the expression matrix $\boldsymbol{X} \in \mathbb{R}^{c \times n}$, we can first employ a multiple-layer GNN to extract the multi-level graph embeddings for all genes:

$$\mathbf{h} = m_g(\mathbf{X}, \mathbf{A}_{tr}), \tag{4}$$

where $\mathbf{h} \in \mathbb{R}^{L \times n \times d}$ denotes all extracted hidden features, $L$ is the layer of GNN, $d$ is the hidden dimension. we employ $\mathbf{h}_i^{(l)} \in \mathbb{R}^{1 \times d}$ to denote the $l$-th layer hidden features for $i$-th gene. In practice, $m_g$ could be implemented as any message-passing models, such as GCN Kipf & Welling (2016), GraphSAGE Hamilton et al. (2017) and GAT Veličković et al. (2018).

To generate further global expert guidance, we compute a global expert score for each candidate pair $(g_i, g_j)$ as:

$$s_{ij} = \mathbf{h}_i^{(L)\top} \mathbf{h}_j^{(L)}, \tag{5}$$

where $\mathbf{h}_i^{(L)}$ and $\mathbf{h}_j^{(L)}$ denote the representations of $g_i$ and $g_j$ at the final GNN layer $L$. Note that the representations $\boldsymbol{h}$ encode multi-scale topological information, where shallow layers focus on local neighborhoods, while deeper layers progressively integrate broader global context, and the expert score $s_{ij}$ thus reflects the most comprehensive structural semantics distilled by the GNN.

**Pairwise Branch**. Following conventional pairwise-based methods, we first extract the embeddings for each candidate pair $(g_i, g_j)$ using their corresponding node embeddings, i.e., $e_i = \boldsymbol{X}_{:,i}$ and $e_j = \boldsymbol{X}_{:,j}$. These embeddings are passed through a multilayer perceptron (MLP) to encode raw features and then fed into a Transformer encoder to generate the initial pairwise representation for downstream fusion:

$$\mathbf{z}_{ij}^{(0)} = m_p(e_i, e_j), \tag{6}$$

where $m_p$ is the pairwise encoding model, $\mathbf{z}_{ij}^{(0)}$ denotes the layer-0 pairwise features of the pair $(g_i, g_j)$, which serve as input to the subsequent multi-scale fusion stage.

**Multi-Scale Fusion**. To fully exploit both expression-level dynamics and global topological context, we design the hierarchical fusion module that injects successive GNN-derived representations $\mathbf{h}$ into the pairwise representation $\mathbf{z}^{(0)}$. Starting from the initial pairwise representations, we perform the following fusion recursively:

$$\mathbf{z}_{ij}^{(l)} = m_f^{(l)}\big(\mathbf{z}_{ij}^{(l-1)}, \mathbf{h}_i^{(l)}, \mathbf{h}_j^{(l)}, \mathbb{I}_{\{l=L\}} \cdot s_{ij}\big), \tag{7}$$

where $m_f^{(l)}$ denotes the $l$-th layer fusion model, $\mathbb{I}_{\{l=L\}}$ is an indicator function that equals 1 if $l = L$ and 0 otherwise, ensuring that $s_{ij}$ is only incorporated at the final fusion layer.

The output $\mathbf{z}_{ij}^{(L)}$ thus integrates fine-grained expression cues with multi-scale global context in a single vector, which is subsequently passed through a sigmoid-activated classifier to produce the predicted probability. The training objective is defined:

$$\mathcal{L}_{\text{ours}} = \sum_{y_{ij} \in E_{\text{tr}}} \big[y_{ij} \log \sigma(\mathbf{z}_{ij}^{(L)}) + (1 - y_{ij}) \log(1 - \sigma(\mathbf{z}_{ij}^{(L)}))\big], \tag{8}$$

where $y_{ij} \in \{0, 1\}$ is the ground-truth label, $\sigma$ denotes the sigmoid-activated classifier, and $\sigma(\mathbf{z}_{ij}^{(L)})$ denotes the model's predicted probability.

## 4 EXPERIMENTS

### 4.1 EXPERIMENTAL SETTING

**Dataset** We evaluate the performance of *scUniGP* on seven scRNA-seq datasets curated by the BEELINE framework Pratapa et al. (2020), covering diverse human and mouse cell types: (i) human embryonic stem cells (hESC), (ii) human hepatocytes (hHEP), (iii) mouse embryonic stem cells (mESC), (iv) mouse dendritic cells (mDC), (v) mouse hematopoietic stem cells with erythroid lineage (mHSC-E), (vi) granulocyte-monocyte lineage (mHSC-GM), and (vii) lymphoid lineage (mHSC-L). Ground-truth GRNs are provided for each dataset based on multiple sources of biological evidence, including: (1) STRING functional interaction networks Szklarczyk et al. (2019), (2)

non-specific ChIP-seq networks Liu et al. (2015); Garcia-Alonso et al. (2019), (3) cell-type-specific ChIP-seq networks Xu et al. (2013b); Moore et al. (2020), and (4) LOF/GOF perturbation-based networks Xu et al. (2013a) for mESC. All datasets are publicly available from GEO: GSE75748 (hESC), GSE81252 (hHEP), GSE98664 (mESC), GSE48968 (mDC), and GSE81682 (mHSC variants). Each scRNA-seq dataset consists of significantly varying transcription factors and either the 500 or 1000 most-variable genes, referred to as TFs500 and TFs1000, respectively. The distribution and statistics of each scRNA-seq dataset with ground-truth networks are shown in the appendix.

**Baselines** We compare *scUniGP* against nine representative GRN inference methods: GE-NIE3 Huynh-Thu et al. (2010), GRNBoost2 Moerman et al. (2019), mutual information (MI) Song et al. (2012), Pearson correlation coefficient (PCC), DeepSEM Shu et al. (2021), GNE KC et al. (2019), GENELink Chen & Liu (2022b), scGREAT Wang et al. (2024), and GNNLink Guo et al. (2023). Among them, GENELink and GNNLink are representative graph-based methods, while GE-NIE3 and scGREAT are typical pairwise-based approaches. MI and PCC are classical co-expression analysis techniques. All models are evaluated using the same expression matrices and candidate TF–target gene pairs to ensure a fair comparison.

**Evaluation Metric** In line with standard practice in GRN inference tasks, we adopt two widely used evaluation metrics: Area Under the Receiver Operating Characteristic Curve (AUROC) and Area Under the Precision-Recall Curve (AUPRC). AUROC reflects the model's overall classification performance, while AUPRC better captures its ability to identify true regulatory interactions under class imbalance, which is common in GRN datasets.

## 4.2 IMPLEMENTATION DETAILS

We follow the preprocessing and splitting protocol in Chen & Liu (2022b), retaining only TF–target interactions and filtering genes by expression variance ($p < 0.01$, Bonferroni-corrected) as in Prat-apa et al. (2020). Positive samples are defined as edges in the ground-truth networks, while all other TF–gene pairs are treated as candidate negatives. Given the sparsity of true regulatory networks, candidate negatives vastly outnumber positives, and some may correspond to undiscovered regulations Thabtah et al. (2020). To address this imbalance, we adopt hard negative sampling (HNS) Radenović et al. (2016), where for each positive TF–gene pair a negative pair with the same TF is uniformly drawn. These hard negatives are more difficult to distinguish from positives and thus provide stronger supervision Zhu et al. (2019a). We randomly assign $67\%$ of positive and HNS pairs to training and validation ($90\%/10\%$), and hold out the remaining $33\%$ for testing, where the proportion of positives approximately matches the network density of the underlying scRNA-seq data. Details of sample distributions and dataset splits are provided in Appendix.

All models are trained on four NVIDIA RTX 4090 GPUs. The global branch of *scUniGP* employs a two-layer GAT or GCN, projecting inputs to 128-dimensional embeddings and then compressing them to 64 dimensions. In the pairwise branch, each TF–gene pair is encoded by two MLPs into 16-dimensional vectors, which are fused via a four-layer Transformer encoder (8 heads, embedding size 1024). Training uses Adam with a learning rate of $5 \times 10^{-6}$, weight decay $1 \times 10^{-5}$, and a step scheduler ($\gamma = 0.999$, step size=10). The model is trained up to 200 epochs with early stopping on validation AUROC (patience=8). Training proceeds in two stages: first, pre-training the GNN on the structural graph; second, jointly fine-tuning the full model with the Transformer while keeping GNN parameters trainable. Further experimental details are provided in the Appendix.

## 4.3 RESULTS AND ANALYSIS

We present *scUniGP* analysis via Q&A, covering benchmark performance and statistical significance, architectural contributions via ablation, and biological interpretability of embeddings.

**Q1. Does *scUniGP* achieve superior GRN inference performance across all benchmarks?**

**A1.** As shown in Table 1, *scUniGP* consistently achieves state-of-the-art GRN inference performance across all evaluated network types(22/22) under the TFs500 benchmark. In terms of AUROC, *scUniGP* attains the highest overall average of 0.911, surpassing the current best-performing method *scGREAT* by 2.13% and outperforming the second-best method *GNNLink* by 4.61%. The performance gap is particularly notable in sparse networks, such as STRING and non-specific ChIP-seq, where *scUniGP* achieves AUROC improvements of 3.08% and 2.63% over *scGREAT*, respectively.

| Network | Method / Dataset | MI† | PCC† | GRN† Boost2 | Deep SEM | GEN† IE3 | GNE | GENE Link | GNN Link | scGR EAT | scUni GP |
|---|---|---|---|---|---|---|---|---|---|---|---|
| STRING | hESC | 0.650 | 0.610 | 0.620 | 0.630 | 0.650 | 0.782 | 0.906 | 0.921 | 0.907 | **0.948** |
| | hHEP | 0.620 | 0.700 | 0.610 | 0.630 | 0.640 | 0.776 | 0.913 | 0.929 | 0.918 | **0.941** |
| | mDC | 0.510 | 0.540 | 0.570 | 0.620 | 0.640 | 0.831 | 0.941 | 0.933 | 0.938 | **0.956** |
| | mESC | 0.670 | 0.640 | 0.610 | 0.630 | 0.640 | 0.802 | 0.926 | 0.924 | 0.934 | **0.951** |
| | mHSC-E | 0.650 | 0.720 | 0.680 | 0.670 | 0.690 | 0.652 | 0.903 | 0.913 | 0.924 | **0.942** |
| | mHSC-GM | 0.720 | 0.810 | 0.780 | 0.740 | 0.780 | 0.736 | 0.910 | 0.905 | 0.920 | **0.937** |
| | mHSC-L | 0.820 | 0.740 | 0.740 | 0.680 | 0.730 | 0.761 | 0.818 | 0.851 | 0.823 | **0.882** |
| Non-specific ChIP-seq | hESC | 0.480 | 0.530 | 0.520 | 0.550 | 0.510 | 0.659 | 0.853 | 0.843 | 0.882 | **0.896** |
| | hHEP | 0.480 | 0.570 | 0.530 | 0.570 | 0.510 | 0.685 | 0.870 | 0.863 | 0.886 | **0.906** |
| | mDC | 0.470 | 0.470 | 0.520 | 0.570 | 0.550 | 0.670 | 0.893 | 0.882 | 0.907 | **0.926** |
| | mESC | 0.540 | 0.550 | 0.540 | 0.550 | 0.550 | 0.649 | 0.887 | 0.861 | 0.879 | **0.928** |
| | mHSC-E | 0.590 | 0.570 | 0.610 | 0.580 | 0.610 | 0.533 | 0.861 | 0.869 | 0.874 | **0.893** |
| | mHSC-GM | 0.650 | 0.610 | 0.640 | 0.600 | 0.660 | 0.562 | 0.851 | 0.861 | 0.880 | **0.882** |
| | mHSC-L | 0.680 | 0.650 | 0.670 | 0.630 | 0.690 | 0.644 | 0.800 | 0.789 | 0.802 | **0.842** |
| LOF/GOF | mESC | 0.680 | 0.550 | 0.650 | 0.640 | 0.650 | 0.778 | 0.854 | 0.871 | 0.888 | **0.891** |
| Specific ChIP-seq | hESC | 0.510 | 0.470 | 0.490 | 0.580 | 0.500 | 0.673 | 0.820 | 0.848 | 0.890 | **0.895** |
| | hHEP | 0.500 | 0.490 | 0.520 | 0.550 | 0.540 | 0.795 | 0.841 | 0.821 | 0.908 | **0.910** |
| | mDC | 0.550 | 0.540 | 0.520 | 0.510 | 0.500 | 0.524 | 0.707 | 0.738 | 0.808 | **0.813** |
| | mESC | 0.530 | 0.510 | 0.530 | 0.500 | 0.500 | 0.808 | 0.882 | 0.885 | 0.930 | **0.941** |
| | mHSC-E | 0.520 | 0.490 | 0.530 | 0.510 | 0.520 | 0.817 | 0.868 | 0.878 | 0.927 | **0.930** |
| | mHSC-GM | 0.490 | 0.540 | 0.500 | 0.530 | 0.530 | 0.831 | 0.894 | 0.892 | 0.928 | **0.937** |
| | mHSC-L | 0.510 | 0.550 | 0.520 | 0.540 | 0.520 | 0.768 | 0.836 | 0.841 | 0.876 | **0.885** |
| —— | **Average** | 0.583 | 0.584 | 0.586 | 0.591 | 0.596 | 0.715 | 0.865 | 0.869 | 0.892 | **0.911** |

Table 1: AUROC performance comparison across different GRN inference methods on benchmark datasets. Methods marked with † are reproduced following the protocol in Wang et al. (2024), while other baselines are our faithful reimplementation of published algorithms under the same experimental settings. All reported results represent the average of the five independent runs with different random seeds, and the variance can be found in the Appendix.

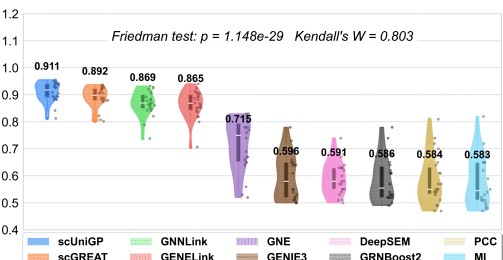

Figure 2: Violin plots illustrate the distribution of AUROC scores for each method. The width of each violin reflects data density, and the internal boxplot summarizes key statistics including the median and quartiles. Scattered dots show individual AUROC values (right of boxes).

Figure 3: Performance comparison of scUniGP and competing methods on gene regulatory network inference tasks. Bar heights represent mean AUPRC values with error bars showing standard deviations. Statistical significance between scUniGP and scGREAT is displayed.

These consistent gains also extend to specific ChIP-seq and perturbation-based networks, suggesting strong generalization capabilities across heterogeneous regulatory conditions. On the larger TFs1000 dataset, *scUniGP* continues to outperform *scGREAT* by a margin of 1.43%, achieving an average AUROC of 0.921. This indicates that the model consistently retains its performance advantage even as the feature space complexity escalates.

AUPRC evaluations further underscore the robustness of *scUniGP*, especially under imbalanced or low-signal scenarios. On TFs500, *scUniGP* achieves an average AUPRC of 0.522, outperforming *scGREAT* (0.463) and the second-best method *GENELink* (0.445). The relative improvements are particularly prominent on the STRING and Non-Specific networks, with gains of approximately 26.0% and 24.7% over *scGREAT*, respectively, echoing the AUROC results. In high-signal con-

| Method | w/o GNN embedding | | w/ GCN embedding | | w/ GAT embedding | |
|--------|--------|--------------|--------|--------------|--------|--------------|
| Network | STRING | Non Specific | STRING | Non Specific | STRING | Non Specific |
| hESC | 0.931 | 0.831 | 0.946 | 0.892 | 0.948 | 0.896 |
| hHEP | 0.935 | 0.893 | 0.949 | 0.901 | 0.941 | 0.906 |
| mDC | 0.952 | 0.894 | 0.956 | 0.907 | 0.956 | 0.926 |
| mESC | 0.933 | 0.878 | 0.947 | 0.901 | 0.951 | 0.928 |
| mHSC-E | 0.922 | 0.875 | 0.936 | 0.855 | 0.942 | 0.893 |
| mHSC-GM | 0.911 | 0.882 | 0.913 | 0.861 | 0.937 | 0.882 |
| mHSC-L | 0.918 | 0.815 | 0.855 | 0.833 | 0.882 | 0.842 |
| Average | 0.918 | 0.867 | 0.929 | 0.879 | 0.937 | 0.896 |

Table 2: Ablation study of the Global Branch in our integral design: AUROC comparison of no GNN, GCN-based, and GAT-based embeddings on TFs500 datasets.

| Method | Early Fusion | | Late Fusion | | Ours | |
|--------|--------|--------------|--------|--------------|--------|--------------|
| Network | STRING | Non Specific | STRING | Non Specific | STRING | Non Specific |
| hESC | 0.923 | 0.828 | 0.932 | 0.902 | 0.948 | 0.896 |
| hHEP | 0.940 | 0.882 | 0.938 | 0.904 | 0.941 | 0.906 |
| mDC | 0.949 | 0.889 | 0.952 | 0.913 | 0.956 | 0.926 |
| mESC | 0.946 | 0.883 | 0.948 | 0.900 | 0.951 | 0.928 |
| mHSC-E | 0.924 | 0.877 | 0.936 | 0.883 | 0.942 | 0.893 |
| mHSC-GM | 0.904 | 0.878 | 0.928 | 0.880 | 0.937 | 0.882 |
| mHSC-L | 0.827 | 0.802 | 0.881 | 0.821 | 0.882 | 0.842 |
| Average | 0.916 | 0.863 | 0.931 | 0.886 | 0.937 | 0.896 |

Table 3: Ablation study of the Multi-Scale Fusion strategy in our integral design: AUROC comparison of early fusion, late fusion, and our proposed multi-scale fusion on TFs500 datasets.

ditions, *scUniGP* remains competitive, achieving AUPRC values comparable to or better than the strongest baselines. Similar trends hold on TFs1000, where *scUniGP* maintains a slight edge over *scGREAT*, further demonstrating its scalability and reliability across diverse transcriptional feature sets. Detailed performance comparisons across all baseline methods are provided in the Appendix.

**Q2. Are the performance improvements of *scUniGP* over baselines statistically significant?**

**A2.** To rigorously assess the significance of *scUniGP*'s performance gains, we evaluated 10 GRN inference methods on the TFs500 datasets across 22 cell types under four types of ground-truth networks. Friedman tests reveal highly significant differences among methods for both AUROC ($\chi^2 = 159.09$, $p = 1.15 \times 10^{-29}$, Kendall's $W = 0.803$) and AUPRC ($\chi^2 = 120.15$, $p = 1.24 \times 10^{-21}$, Kendall's $W = 0.607$), while Nemenyi post-hoc tests confirm that *scUniGP* consistently ranks first and differs significantly from most baselines. To further validate these findings, we performed paired tests against the strongest baseline *scGREAT*, showing that *scUniGP* achieves significantly higher AUROC (Cohen's $d = 1.16$) and AUPRC (Cohen's $d = 0.72$), with particularly strong gains on STRING and Non-Specific networks. Figures 2 and 3 provide visual support for these findings: the former depicts AUROC distributions with violin–scatter plots, while the latter presents AUPRC comparisons using bar charts with error bars. Together, they show that *scUniGP*'s improvements are both consistent and statistically significant. Further statistical analyses, dataset-specific results, and TFs1000 evaluations are provided in the Appendix.

**Q3: What is the contribution of global branch and multiscale fusion in *scUniGP*?**

**A3:** To elucidate the contributions of key architectural components in *scUniGP*, we conduct comprehensive ablation experiments on seven TFs500 datasets using both the STRING and Non-Specific cell-type networks. We first examine the effect of GNN-based gene embeddings by comparing three variants: (i) without GNN embeddings, (ii) with GCN-based multiscale embeddings, and (iii) with GAT-based multiscale embeddings. As shown in Table 2, the average AUROC improves from 0.918 (w/o GNN) to 0.929 with GCN and 0.937 with GAT on STRING, and from 0.867 to 0.879 and 0.896, respectively, on Non-Specific networks. GCN yields improvements on 10 out of 14 datasets,

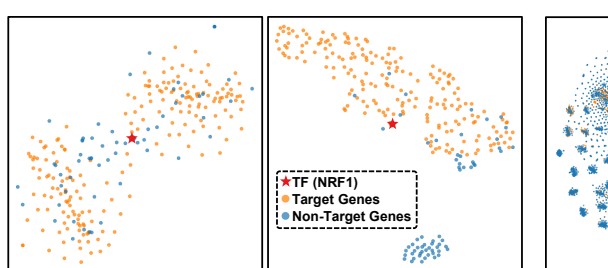 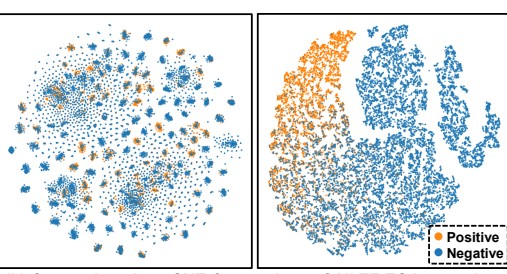

(a) TF-NRF1 t-SNE Visualization of Input and Output Embeddings   (b) Comprehensive t-SNE Comparison of All TF-TG Interactions

Figure 4: t-SNE visualization of transcription factor-target gene interactions in both input and output embedding spaces. (a) shows TF NRF1 and its target/non-target genes, with the left panel representing input embeddings based on raw gene expression data and the right panel showing output embeddings from the model's penultimate layer. (b) presents visualizations of all TF-target interactions, with the left panel showing input embeddings based on flattened raw gene expression data and the right panel displaying output embeddings from the model's penultimate layer.

whereas GAT consistently enhances performance across all cases and achieves the best overall results, demonstrating the effectiveness of GNN-based embeddings.

Next, using the GAT-embedded model, we evaluate three fusion strategies: early fusion (feature concatenation before the Transformer encoder), late fusion (decision-level ensemble), and our proposed multiscale (layer-wise) fusion. As shown in Table 3, multiscale fusion achieves the best overall AUROC, surpassing early and late fusion by 2.24% and 0.64% on STRING, and by 3.82% and 1.13% on Non-Specific networks. Notably, early fusion even underperforms the no-GNN variants, indicating that embeddings learned at deeper GNN layers capture progressively richer features and thus require layer-wise integration with the Pairwise Branch. These results confirm that GNN-based embeddings and multiscale fusion are essential to the robust performance of *scUniGP*.

**Q4: Does *scUniGP* learn biologically meaningful representations of gene regulation?**

**A4:** To assess the representational effectiveness of *scUniGP*, we conducted two t-SNE visualization experiments to examine whether the model can learn biologically meaningful and discriminative embeddings that separate true TF–target gene (TG) pairs from non-target pairs. First, for the TF *NRF1* in the mESC dataset (Cell-type-Specific network), input embeddings—constructed by flattening raw expression vectors—show substantial overlap between TG and non-TG pairs. In contrast, output embeddings from the penultimate layer, which integrate gene expression, multiscale GNN interactions, and expert attention, exhibit clear clustering with NRF1 centered among its targets (Figure 4a), demonstrating improved intra-class cohesion and inter-class separation. In the second experiment, extending to all TF–TG pairs across datasets, output embeddings consistently enhance class separation relative to the input space (Figure 4b), indicating that *scUniGP* transforms high-dimensional sparse expression data into a more discriminative feature space. Additional TF-specific and global embedding visualizations are provided in the Appendix.

## 5   CONCLUSION

We proposed *scUniGP*, a unified framework for GRN inference that integrates global graph topology with local TF–gene interactions via hierarchical multi-scale fusion. Across diverse benchmarks, *scUniGP* consistently outperforms state-of-the-art methods, and ablations verify the effectiveness of the global branch and fusion strategy. These results demonstrate its ability to mitigate annotation sparsity and overfitting, yielding robust and biologically meaningful representations.

Supervised GRN inference lacks reliable negatives, though our uniform sampling reduces false negatives. The joint design may also trade some efficiency compared with graph- or pairwise-only methods. In future work, we plan to extend *scUniGP* with richer modalities (e.g., epigenetics, sequence, protein structure) for more comprehensive modeling. Moreover, while large-scale biological foundation models have shown limited impact on GRN-level tasks, we aim to explore their integration within *scUniGP* to further enhance model generalization, scalability, and interpretability.

## REPRODUCIBILITY STATEMENT

We have made extensive efforts to ensure the reproducibility of our work. Detailed descriptions of dataset preprocessing and splitting protocols are provided in Appendix A, while model architectures, hyperparameter settings, and training strategies are reported in Appendix B. Comprehensive experimental results, including performance comparisons, visualizations, and runtime analysis, are presented in Appendix C. All algorithms and theoretical formulations are clearly specified in the main text and Appendix, with assumptions and derivations explicitly stated where applicable. Furthermore, the complete source code used for training and evaluation is included in the supplemental materials, enabling full reproduction of our experiments.

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

APPENDIX

This appendix provides additional details to supplement the main paper, including: **method details**, where we present a comprehensive formulation of Eq. (4) and Eq. (6) from the main text; **dataset details**, where we describe each dataset used in our experiments, including statistics and preprocessing procedures; **implementation details**, where we elaborate on the adaptations of existing models and the configurations of downstream tasks; and **experimental details**, where we provide supplementary results—such as variances and significance tests omitted from the main paper—along with extended discussions and analyses that further support our benchmark findings.

## A  METHOD

**Overview.** SCUNIGP is a hybrid framework that integrates global regulatory topology with fine-grained expression dynamics. In the Appendix, we provide detailed formulations for each component, focusing on the Global Branch and Pairwise Branch, which correspond to Eqs. (4) and (6).

**Global Branch (Eq. (4)).** We adopt a general message-passing GNN to extract multi-scale topological features from the prior regulatory graph $\mathbf{A}_{tr}$ and the expression matrix $\mathbf{X}$, as formulated in Eq. (4) of the main paper. The layer-wise propagation follows a unified formulation:

$$\mathbf{h}_i^{(l)} = \mathrm{AGG}^{(l)} \left( \left\{ \phi^{(l)}(\mathbf{h}_i^{(l-1)}, \mathbf{h}_j^{(l-1)}, \mathbf{e}_{ij}) \mid j \in \mathcal{N}(i) \right\} \right), \quad l = 1, \dots, L \tag{a}$$

where $\mathbf{h}_i^{(l)} \in \mathbb{R}^d$ denotes the embedding of gene $g_i$ at layer $l$, $\mathcal{N}(i)$ is the set of its neighbors, $\mathbf{e}_{ij}$ represents optional edge features, $\phi^{(l)}(\cdot)$ is a learnable message function, and $\mathrm{AGG}^{(l)}$ is a layer-specific aggregation operator such as mean, sum, or attention. Each layer optionally includes a nonlinear activation (e.g., ReLU) and residual connection to improve gradient flow and training stability. Our framework supports standard GNN instantiations such as GCN Kipf & Welling (2017), where messages are aggregated via normalized adjacency, and GAT Veličković et al. (2018), which employs attention-based neighbor weighting to enhance expressive power.

The hierarchical embeddings $\mathbf{h}^{(1:L)}$ capture multi-scale topological context: shallow layers encode local neighborhood information, intermediate layers integrate broader connectivity patterns, and the final layer aggregates global structure. For each candidate TF–target pair $(g_i, g_j)$, the global expert score is computed based on the final-layer embeddings, as defined in Eq. (5) of the main paper.

**Pairwise Branch (Eq. (6)).** For each candidate pair $(g_i, g_j)$, we first extract their expression vectors $\mathbf{x}_i, \mathbf{x}_j \in \mathbb{R}^c$ from the matrix $\mathbf{X}$. These vectors are projected into a shared latent space via a linear transformation $\ell(\cdot)$, and combined with learnable positional embeddings $\mathbf{p}_0, \mathbf{p}_1 \in \mathbb{R}^d$, which are initialized as simple 0/1 vectors to distinguish the roles of TF and target. The resulting sequence is then jointly encoded using a Transformer encoder:

$$\mathbf{z}_{ij}^{(0)} = \mathrm{TransformerEncoder}\left([\ell(\mathbf{x}_i) + \mathbf{p}_0 \ ; \ \ell(\mathbf{x}_j) + \mathbf{p}_1]\right) \in \mathbb{R}^{d_t} \tag{b}$$

This representation $\mathbf{z}_{ij}^{(0)}$ captures cell-contextualized expression dependencies between the TF $g_i$ and target $g_j$, serving as the initial input for the subsequent multi-scale fusion. During fusion, these pairwise embeddings are concatenated with the corresponding GNN-derived multi-scale embeddings from the Global Branch to fully integrate local expression patterns with global topological context.

**Summary.** In summary, SCUNIGP integrates multi-scale GNN embeddings and expert scores from the Global Branch with fine-grained, context-specific pairwise embeddings via concatenation in the Multi-scale Fusion module, producing a unified representation for accurate TF–target inference; the overall training procedure is detailed in Algorithm 1.

## B  DATASET

We adopt seven benchmark scRNA-seq datasets provided by the BEELINE pipeline, covering a range of biological systems and experimental conditions. To provide supervision for GRN inference, we utilize four types of ground-truth regulatory networks: (1) **STRING networks**, derived

---

**Algorithm 1** Training Procedure of SCUNIGP

---

1: **Input:** Expression matrix $\boldsymbol{X} \in \mathbb{R}^{c \times n}$; adjacency matrix $\mathbf{A}_{tr} \in \{0,1\}^{n \times n}$; edge labels $y_{ij}$ for $(g_i, g_j) \in E_{tr}$

2: **Output:** Trained model for gene regulatory inference

3: **Global Branch**: Apply multi-layer GNN $m_g$ to extract multi-scale gene features:

$$\mathbf{h} \leftarrow m_g(\boldsymbol{X}, \mathbf{A}_{tr}) \quad \text{where } \mathbf{h} \in \mathbb{R}^{L \times n \times d}$$

4: **for all** $(g_i, g_j) \in E_{tr}$ **do**

5:     Extract top-layer embeddings: $\mathbf{h}_i^{(L)}, \mathbf{h}_j^{(L)}$

6:     Compute expert score: $s_{ij} \leftarrow \mathbf{h}_i^{(L)\top} \mathbf{h}_j^{(L)}$

7: **end for**

8: **for all** $(g_i, g_j) \in E_{tr}$ **do**

9:     Extract expression vectors: $e_i \leftarrow \boldsymbol{X}_{:,i}, e_j \leftarrow \boldsymbol{X}_{:,j}$

10:     **Pairwise Branch**: Encode pair via Transformer:

$$\mathbf{z}_{ij}^{(0)} \leftarrow m_p(e_i, e_j)$$

11:     **for** $l = 1$ to $L$ **do**

12:       **if** $l = L$ **then**

13:         Fuse with final graph embeddings and expert score:

$$\mathbf{z}_{ij}^{(l)} \leftarrow m_f^{(l)}(\mathbf{z}_{ij}^{(l-1)}, \mathbf{h}_i^{(l)}, \mathbf{h}_j^{(l)}, s_{ij})$$

14:       **else**

15:         Fuse with intermediate graph features:

$$\mathbf{z}_{ij}^{(l)} \leftarrow m_f^{(l)}(\mathbf{z}_{ij}^{(l-1)}, \mathbf{h}_i^{(l)}, \mathbf{h}_j^{(l)})$$

16:       **end if**

17:     **end for**

18:     Predict score: $\hat{y}_{ij} \leftarrow \sigma(\mathbf{z}_{ij}^{(L)})$

19:     Compute loss:

$$\mathcal{L}_{ij} \leftarrow y_{ij} \log \hat{y}_{ij} + (1 - y_{ij}) \log(1 - \hat{y}_{ij})$$

20: **end for**

21: **Update** model parameters via total loss:

$$\mathcal{L}_{\text{ours}} \leftarrow \sum_{(g_i, g_j) \in E_{tr}} \mathcal{L}_{ij}$$

---

from protein-protein interaction databases; (2) **non-cell-type-specific ChIP-seq networks(Non-Specific)**, built from TF binding profiles aggregated across diverse contexts; (3) **cell-type-specific ChIP-seq (Specific)** networks, offering high-resolution, context-specific TF–target interactions; and (4) **LOF/GOF networks**, based on experimentally validated perturbation-derived causal interactions. Among these, the Specific networks typically include more positive TF–target pairs, resulting in more balanced datasets. In contrast, STRING and Non-Specific networks are much sparser and often yield highly imbalanced classification settings. Each dataset is evaluated under two scales: **TFs500** and **TFs1000**, referring to the top 500 and 1000 TFs respectively ranked by expression and variability. All results are averaged over five independent trials with different random seeds to ensure statistical robustness and reliability.

**Dataset Construction and Splitting.** To construct a binary classification dataset, we treat each TF–target pair as an instance. Regulatory relationships annotated in the ground-truth networks are taken as positive samples, while all remaining TF–gene pairs are treated as candidate negatives Marbach et al. (2012). Since true regulatory networks are extremely sparse, the total number of candidate negative pairs greatly exceeds that of positives, and some negatives may correspond to undiscovered regulatory interactions De Smet & Marchal (2010); Blatti et al. (2015); Yang et al. (2022); Zhu et al.

| Dataset | Genes | Cells | STRING | | | | | Non-Specific | | | | |
|---|---|---|---|---|---|---|---|---|---|---|---|---|
| | | | TFs | Targets | Positive | Density | Trainingsets | TFs | Targets | Positive | Density | Trainingsets |
| hESC | 910 | 758 | 343 | 511 | 4257 | 0.024 | 208614 | 283 | 753 | 3441 | 0.016 | 172153 |
| hHEP | 948 | 425 | 409 | 646 | 7523 | 0.028 | 259147 | 322 | 825 | 4129 | 0.015 | 204039 |
| mDC | 821 | 383 | 264 | 479 | 4815 | 0.038 | 144820 | 250 | 634 | 3067 | 0.019 | 137156 |
| mESC | 1120 | 421 | 495 | 638 | 7762 | 0.024 | 370740 | 516 | 890 | 6893 | 0.015 | 386511 |
| mHSC-E | 704 | 1071 | 156 | 291 | 1371 | 0.029 | 73346 | 144 | 442 | 1425 | 0.022 | 67712 |
| mHSC-GM | 632 | 889 | 92 | 201 | 748 | 0.040 | 38827 | 82 | 297 | 743 | 0.030 | 34615 |
| mHSC-L | 560 | 847 | 39 | 70 | 137 | 0.048 | 14573 | 35 | 164 | 279 | 0.048 | 13081 |

| Dataset | Genes | Cells | Cell-type Specific | | | | | LOF/GOF | | | | |
|---|---|---|---|---|---|---|---|---|---|---|---|---|
| | | | TFs | Targets | Positive | Density | Trainingsets | TFs | Targets | Positive | Density | Trainingsets |
| hESC | 910 | 758 | 34 | 815 | 4545 | 0.164 | 20677 | - | - | - | - | - |
| hHEP | 948 | 425 | 30 | 874 | 9939 | 0.379 | 19002 | - | - | - | - | - |
| mDC | 821 | 383 | 20 | 443 | 756 | 0.085 | 10969 | - | - | - | - | - |
| mESC | 1120 | 421 | 88 | 977 | 29613 | 0.345 | 65895 | 34 | 774 | 4169 | 0.158 | 25459 |
| mHSC-E | 704 | 1071 | 23 | 691 | 11557 | 0.578 | 13632 | - | - | - | - | - |
| mHSC-GM | 632 | 889 | 22 | 618 | 7364 | 0.543 | 9280 | - | - | - | - | - |
| mHSC-L | 560 | 847 | 16 | 525 | 4398 | 0.525 | 5976 | - | - | - | - | - |

Table 4: Statistics of TFs-500 ground-truth networks for seven datasets.

| Dataset | Genes | Cells | STRING | | | | | Non-Specific | | | | |
|---|---|---|---|---|---|---|---|---|---|---|---|---|
| | | | TFs | Targets | Positive | Density | Trainingsets | TFs | Targets | Positive | Density | Trainingsets |
| hESC | 1410 | 758 | 351 | 695 | 5149 | 0.021 | 331058 | 292 | 1138 | 4617 | 0.014 | 275435 |
| hHEP | 1448 | 425 | 414 | 874 | 9003 | 0.024 | 401011 | 332 | 1331 | 5351 | 0.013 | 321581 |
| mDC | 1321 | 383 | 273 | 664 | 5898 | 0.032 | 241221 | 254 | 969 | 3918 | 0.016 | 224444 |
| mESC | 1620 | 421 | 499 | 785 | 8479 | 0.021 | 540893 | 522 | 1214 | 8030 | 0.013 | 565848 |
| mHSC-E | 1204 | 1071 | 161 | 413 | 1826 | 0.027 | 129653 | 147 | 674 | 1960 | 0.020 | 118364 |
| mHSC-GM | 1132 | 889 | 100 | 344 | 1311 | 0.037 | 75698 | 88 | 526 | 1358 | 0.029 | 66621 |
| mHSC-L | 692 | 847 | 40 | 81 | 154 | 0.045 | 18487 | 37 | 192 | 317 | 0.043 | 17096 |

| Dataset | Genes | Cells | Cell-type Specific | | | | | LOF/GOF | | | | |
|---|---|---|---|---|---|---|---|---|---|---|---|---|
| | | | TFs | Targets | Positive | Density | Trainingsets | TFs | Targets | Positive | Density | Trainingsets |
| hESC | 1410 | 758 | 34 | 1260 | 7084 | 0.165 | 32065 | - | - | - | - | - |
| hHEP | 1448 | 425 | 31 | 1331 | 15558 | 0.377 | 30026 | - | - | - | - | - |
| mDC | 1321 | 383 | 21 | 684 | 1193 | 0.082 | 18556 | - | - | - | - | - |
| mESC | 1620 | 421 | 89 | 1385 | 42795 | 0.347 | 96460 | 34 | 1098 | 5742 | 0.154 | 36848 |
| mHSC-E | 1204 | 1071 | 29 | 1177 | 21975 | 0.566 | 26565 | - | - | - | - | - |
| mHSC-GM | 1132 | 889 | 23 | 1089 | 14135 | 0.561 | 17406 | - | - | - | - | - |
| mHSC-L | 692 | 847 | 16 | 640 | 5180 | 0.507 | 7392 | - | - | - | - | - |

Table 5: Statistics of TFs-1000 ground-truth networks for seven datasets.

(2019b). This extreme imbalance can hinder the model's ability to focus on learning positive regulations effectively. To address this issue, we adopt the hard negative sampling (HNS) strategy Chum (2017); Chen & Liu (2022a); Radenović et al. (2016); Thabtah et al. (2020); Zhu et al. (2019a) as in GENELink Chen & Liu (2022b). For each positive TF–target pair, a negative pair involving the same TF is uniformly sampled from the pool of candidate negatives. These hard negatives are challenging to distinguish from positives because they share the same TF and exhibit similar expression patterns, thereby providing stronger supervision and encouraging the model to focus on fine-grained discriminative features Zhu et al. (2019b); Yang et al. (2022).

Following Chen & Liu (2022b), we perform stratified splitting per TF. Specifically, for each TF, $2/3$ of its positive and selected hard negative samples are randomly assigned to training and validation sets in a 9:1 ratio, and the remaining $1/3$ is held out for independent testing. If a TF has only one positive target, it is randomly assigned to either the training or testing set. For TFs with two positives, one is used for training and the other for testing. For TFs with more than two positives, the samples are split according to the 67%/6.7%/23.3% ratio for training/validation/testing, and negative samples are partitioned in the same proportion. This TF-specific splitting avoids information leakage across evaluation stages. All datasets are preprocessed by retaining only TF–target interactions and filtering genes based on expression variance ($p < 0.01$, Bonferroni-corrected) following Pratapa et al. (2020). Table 4 and Table 5 summarize the statistics of the resulting datasets. Overall, the proportion of positive samples approximately reflects the sparsity of the ground-truth regulatory network within each scRNA-seq dataset.

## C  IMPLEMENTATIONS

In this section, we present detailed implementation strategies for our model SCUNIGP, including the global structural embedding branch and the pairwise expression-guided prediction module. To ensure consistency and reproducibility, all components are trained under a unified experimental protocol with standardized optimization settings, regularization strategies, and evaluation procedures.

**Global Branch: GNN Embedding Generation.** To provide structural priors for downstream prediction, we pre-train a two-stage GNN architecture (either GAT or GCN) on the ground-truth TF–target interaction graph. Both GNN models consist of three hidden layers with dimensions 128, 64, and 32, producing final embeddings of dimension 16. For GAT, we use 3 attention heads per layer with LeakyReLU activation and concatenation; for GCN, symmetric normalization with ReLU activation is applied. Dropout with a rate of 0.01 is used between layers. The input graph is constructed from positive TF–target pairs in the training set, with undirected edges and without self-loops. Training uses the Adam optimizer with a learning rate of $3 \times 10^{-3}$, binary cross-entropy loss, batch size 256, early stopping after 5 epochs of no improvement, and a maximum of 30 epochs. After training, we extract four types of embeddings: TF-specific and target-specific embeddings (16-dimensional), as well as Level-1 and Level-2 gene embeddings (128- and 64-dimensional), which encode local and global graph structure. These embeddings are stored for fusion with expression-based features in the main model.

**Pairwise Branch: GRN Prediction.** The main prediction module adopts a dual-path fusion architecture that integrates expression features with GNN-derived structural embeddings for TF–target interaction classification. The expression path first projects standardized expression profiles through a linear transformation, followed by a 4-layer Transformer encoder with 8 attention heads and embedding size 1024. Positional encoding is applied to capture cell-specific dependencies. In parallel, the graph path maps multi-scale GNN embeddings into the same latent space and concatenates them with the transformer output. The combined features are fed into a three-layer classifier with residual connections, batch normalization, and PReLU activation. GNN prediction scores are additionally incorporated during the final decision stage for late fusion. Training uses binary cross-entropy loss with the Adam optimizer, a learning rate of $5 \times 10^{-6}$, weight decay $1 \times 10^{-5}$, dropout rate 0.2, and L2 regularization with $\lambda = 0.01$. Batch size is set to 512, with early stopping after 8 epochs of no improvement and a maximum of 200 epochs. To incorporate local context, up to five neighboring gene pairs are sampled from the graph during training. Model selection is based on validation AUROC, and both the best and average performance on the test set are reported.

## D  EXPERIMENTS

In this section, we systematically evaluate the performance of SCUNIGP on benchmark scRNA-seq datasets and associated regulatory networks. We compare TF–target interaction inference under two settings, TFs500 and TFs1000, using AUROC and AUPRC as primary evaluation metrics, with results averaged over three independent runs. In addition to predictive performance, we report the overall computational efficiency of SCUNIGP in terms of runtime. To further interpret the model, we also visualize the learned TF–target embeddings using t-SNE, illustrating how positive and negative interactions are separated in the latent space and how the model enhances representation discriminability during training.

**Overall Performance Across All Datasets.** We begin by providing a comprehensive comparison of model performance across all datasets using AUROC and AUPRC heatmaps, as shown in Figure 5-Figure 8. Each heatmap reports scores under both TFs500 and TFs1000 settings for all competing methods across the four types of regulatory networks. In terms of AUROC (Figure 5 and Figure 6), SCUNIGP achieves the best performance on 40 out of 42 datasets, consistently outperforming baseline methods across STRING, Specific, Non-Specific and LOF/GOF networks. This reflects strong generalization across both broad and context-specific regulatory priors. For AUPRC (Figure 7 and Figure 8), SCUNIGP remains the top-performing model on 33 of 42 datasets, showing superior capability in capturing true regulatory interactions under class-imbalanced settings.

**Performance Comparison via Bar and Violin Plots.** To comprehensively assess model performance across various datasets and network conditions, we present bar plots (Figures 9–12) and violin plots (Figures 13–16). The bar plots illustrate the average AUROC and AUPRC scores of each

| Running time | scUniGP | scGREAT | GNNLink | GENELink | GNE | CNNC | DeepSEM | PCC | MI |
|---|---|---|---|---|---|---|---|---|---|
| TF+500 genes | 18m21s | 15m10s | 2m01s | 2m10s | 27m20s | 10h54s | 2m08s | 15s | 16s |
| TF+1000 genes | 41m08s | 37m40s | 2m45s | 3m50s | 52m08s | 38h14s | 3m28s | 28s | 26s |

Table 6: Comparison of running time between SCUNIGP and other methods.

method under TFs500 and TFs1000 settings across four types of ground-truth networks: STRING, Non-Specific, LOF/GOF perturbation, and Specific. Mean values are annotated above each bar, with error bars indicating the variation across datasets. The violin plots visualize the full score distributions for each method, capturing both central tendencies and variance. Wider regions reflect higher score density, with internal box plots summarizing median and quartiles, and overlaid dots marking individual values. Statistical significance tests are annotated to highlight reliable differences.

**t-SNE Visualizations of TF–Target Representations.** To gain insights into the internal representations learned by SCUNIGP, we visualize the embedding space of TF–target interactions using t-SNE. Figures 17–19 illustrate four representative TFs from the mESC dataset under the Specific network, highlighting the spatial organization of positive and negative gene pairs. Each figure shows the input embedding from expression features (left) and the transformed output embedding from the penultimate model layer (right). We observe that the model effectively enhances separation between true and false targets during training. Figures 20– 22 further present a global view of all TF–target interactions from the hESC,hHEP,mHSC-E and mHSC-GM datasets, respectively. These plots show the distribution of positive and negative samples before and after feature transformation, revealing that SCUNIGP consistently learns more compact and discriminative representations in the latent space.

**Computational Efficiency Analysis.** To assess computational efficiency, we report the average runtime of each method on all STRING datasets, which contain the largest number of training samples among the benchmark networks (Table 6). All experiments were conducted on a workstation running Ubuntu 22.04.4 LTS, equipped with an Intel(R) Xeon(R) Gold 6133 CPU (40 physical cores, 80 threads, 2.50 GHz base frequency up to 3.00 GHz), 64 GB RAM, and an NVIDIA GeForce RTX 4090 GPU. On the TFs500 setting, SCUNIGP required approximately 18 minutes, while on the TFs1000 the runtime increased to around 41 minutes. This demonstrates competitive scalability relative to other deep learning–based approaches. In particular, compared to the state-of-the-art baseline sc-GREAT (15 minutes for TFs500 and 38 minutes for TFs1000), SCUNIGP sacrifices only about 8.5% computational efficiency but achieves an average improvement of 1.4 points in predictive performance (AUROC/AUPRC). By contrast, methods such as GENELink and GNNLink are faster (2–4 minutes) but come at a significant cost in accuracy, while traditional statistical baselines (PCC and MI) finish within seconds yet lack competitive inference quality. We also note that some models, such as CNNC and GNE, show prohibitive runtimes: CNNC requires nearly 11 hours for TFs500 and more than 38 hours for TFs1000, while GNE needs over 25–50 minutes. These results highlight that SCUNIGP achieves a favorable trade-off between runtime and predictive performance, maintaining efficiency while substantially outperforming both deep learning and classical baselines.

# E  DISCUSSION

A central challenge in GRN inference is the sparsity and imbalance of scRNA-seq data, which limits the effectiveness of conventional AI-based approaches. By integrating structural priors with context-specific expression patterns, SCUNIGP achieves robust and consistent performance across diverse datasets and regulatory priors, highlighting the benefit of combining global and local perspectives.

Despite these strengths, our framework currently relies solely on scRNA-seq input, potentially overlooking complementary regulatory signals from other modalities such as chromatin accessibility or DNA methylation. In future work, we plan to extend SCUNIGP into a multimodal framework and leverage large-scale pretrained models (e.g., scGPT Cui et al. (2024)) to enhance generalization across cell types and conditions, thereby advancing the scalability and biological interpretability of GRN inference.

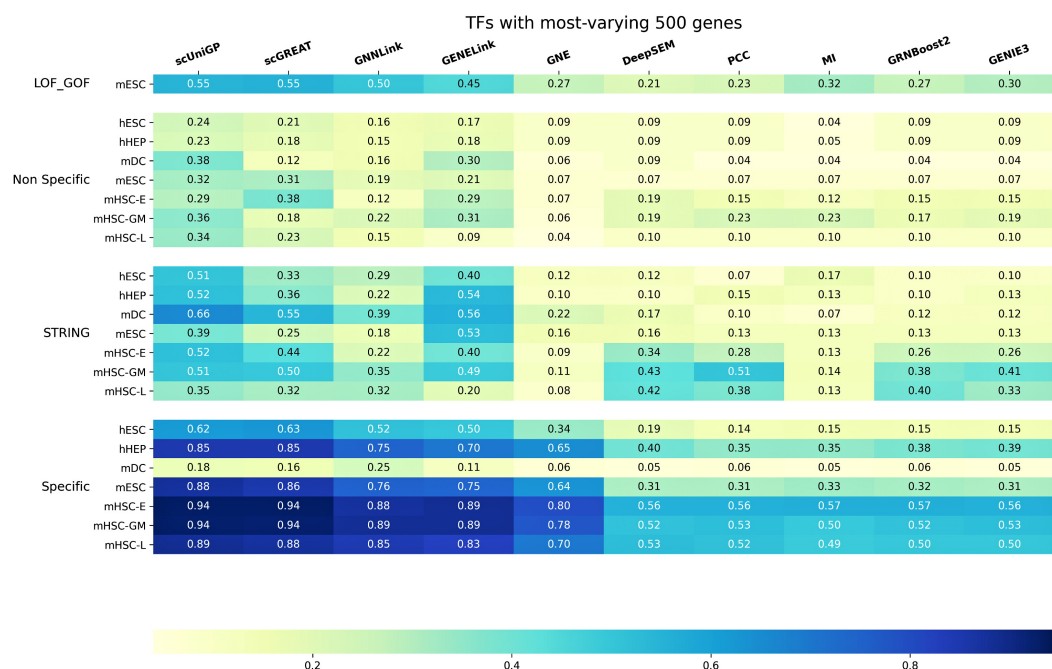

Figure 5: AUROC heatmap comparing model performance across all datasets on TFs500. Each cell represents the mean AUROC for a model–dataset–network combination.

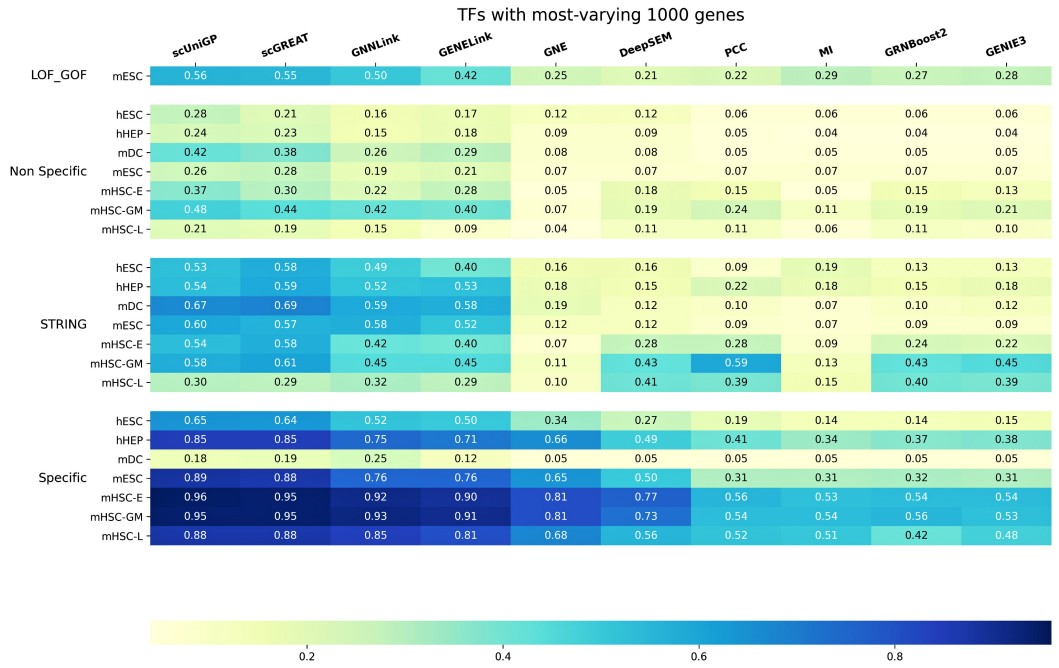

Figure 6: AUROC heatmap comparing model performance across all datasets on TFs1000. Each cell represents the mean AUROC for a model–dataset–network combination.

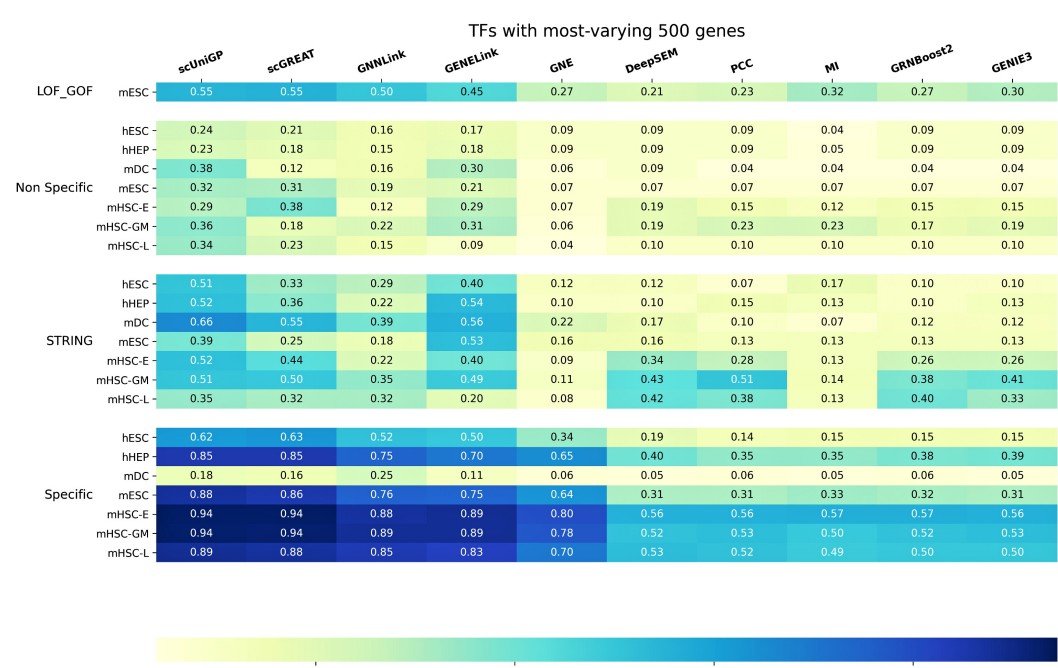

Figure 7: AUPRC heatmap comparing model performance across all datasets on TFs500. Each cell represents the mean AUPRC for a model–dataset–network combination.

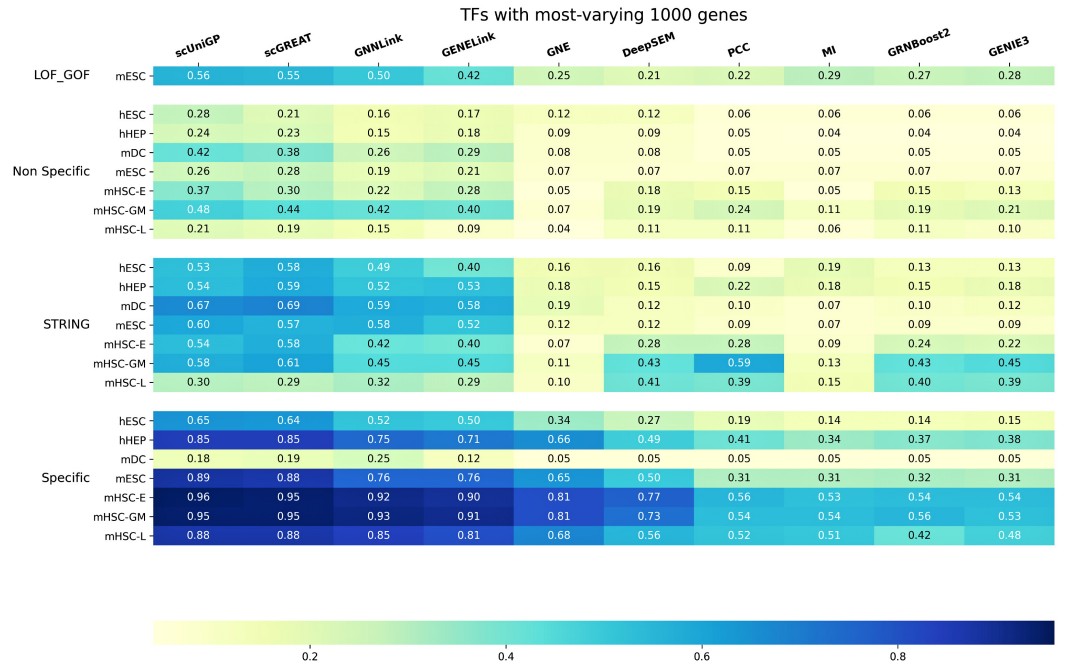

Figure 8: AUPRC heatmap comparing model performance across all datasets on TFs1000. Each cell represents the mean AUPRC for a model–dataset–network combination.

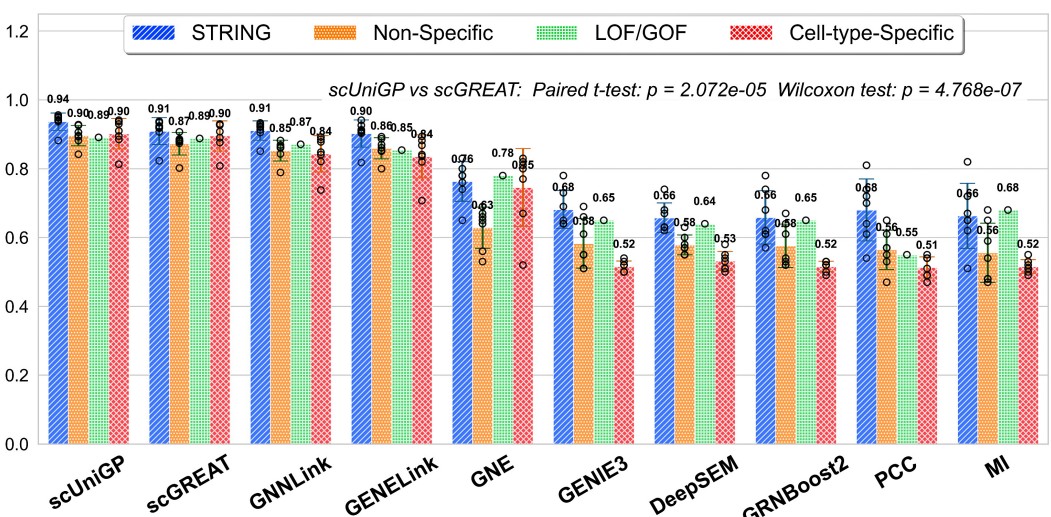

Figure 9: Bar plot of average AUROC scores under four network types on the TFs500, with error bars indicating normalized standard deviation across cell types and means annotated above each bar.

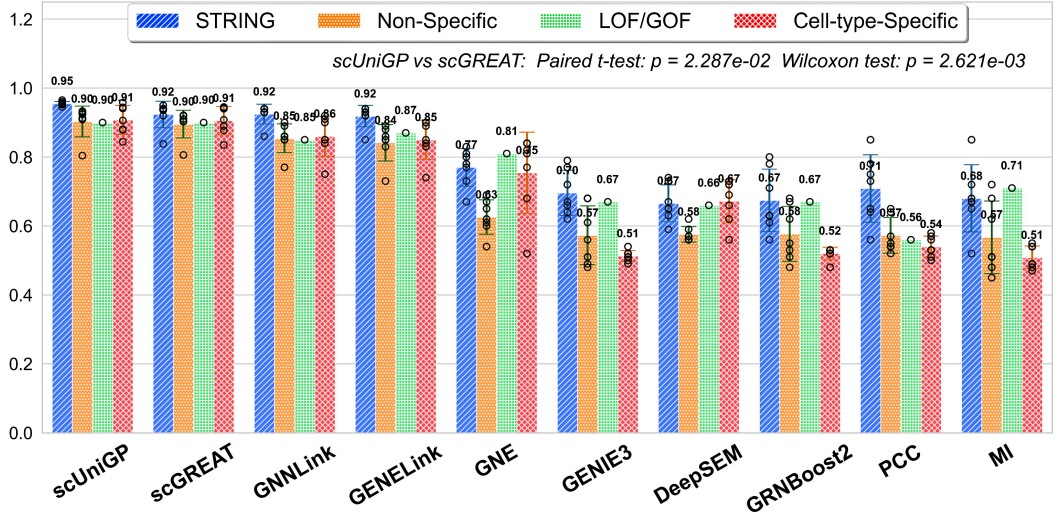

Figure 10: Bar plot of average AUROC scores under four network types on the TFs1000, with error bars indicating normalized standard deviation across cell types and means annotated above each bar.

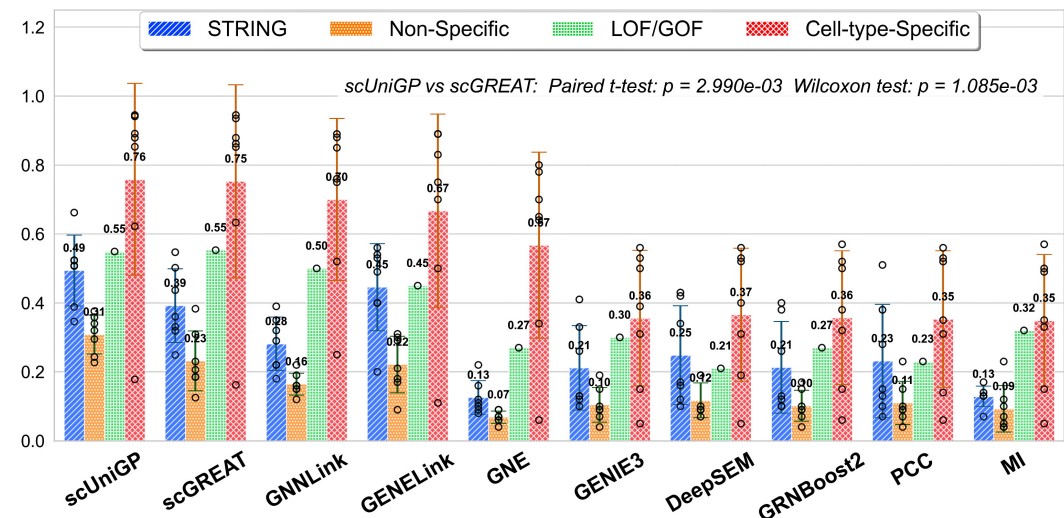

Figure 11: Bar plot of average AUPRC scores under four network types on the TFs500, with error bars indicating normalized standard deviation across cell types and means annotated above each bar.

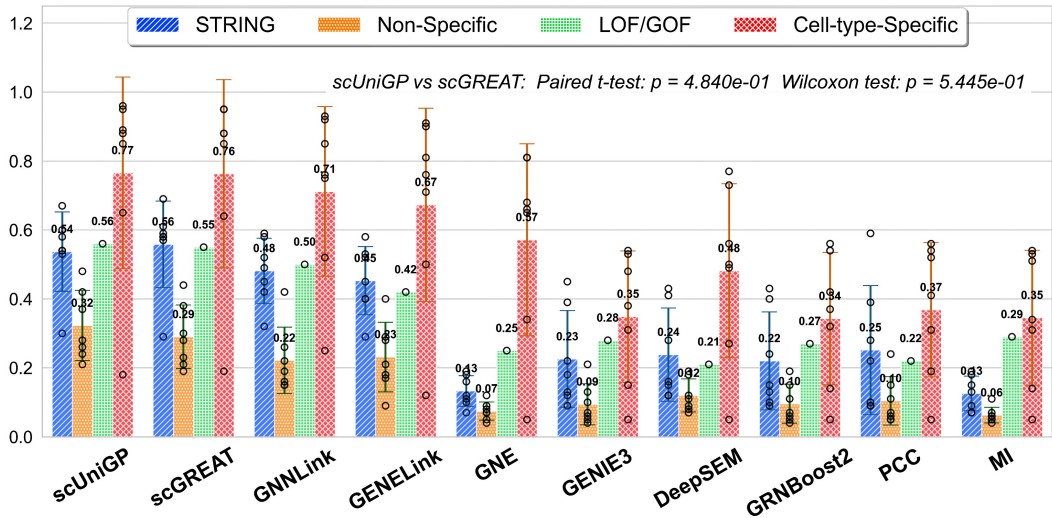

Figure 12: Bar plot of average AUPRC scores under four network types on the TFs1000, with error bars indicating normalized standard deviation across cell types and means annotated above each bar.

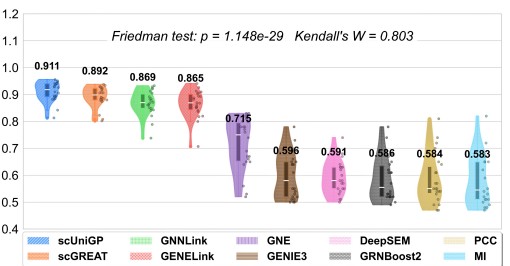

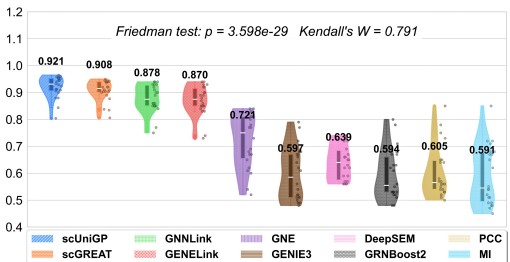

Figure 13: Violin plot of AUROC distributions across all dataset-network combinations for TFs500, with boxplots showing median and IQR, overlaid points, and significance markers.

Figure 14: Violin plot of AUROC distributions across all dataset-network combinations for TFs1000, with boxplots showing median and IQR, overlaid points, and significance markers.

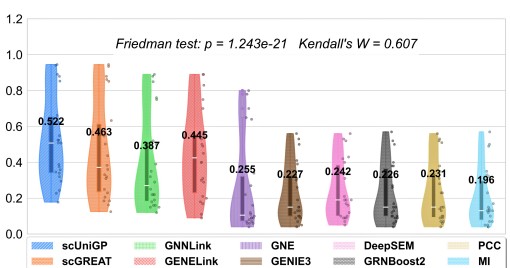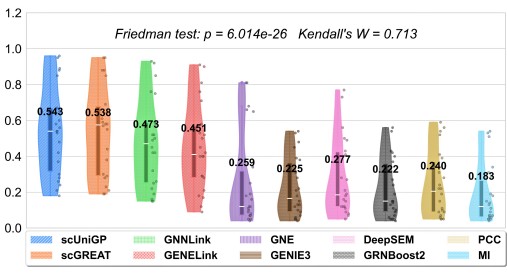

Figure 15: Violin plot of AUPRC distributions across all dataset-network combinations for TFs500, with boxplots showing median and IQR, overlaid points, and significance markers.

Figure 16: Violin plot of AUPRC distributions across all dataset-network combinations for TFs1000, with boxplots showing median and IQR, overlaid points, and significance markers.

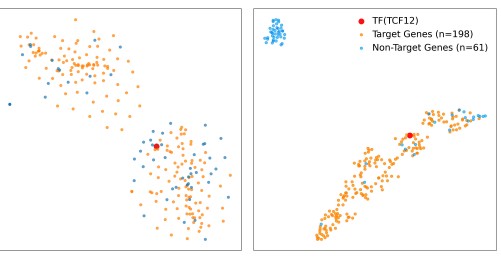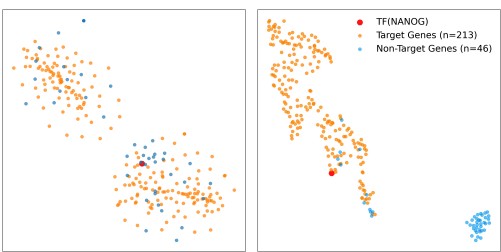

Figure 17: t-SNE visualization for TF TCF12. Left: Input embeddings based on raw expression. Right: Model-learned output embeddings. Target genes form tighter clusters post-training.

Figure 18: t-SNE visualization for TF NANOG. Left: Input embeddings based on raw expression. Right: Model-learned output embeddings. Target genes form tighter clusters post-training.

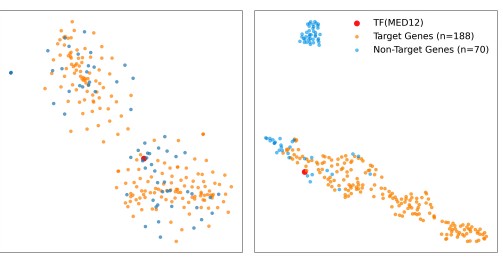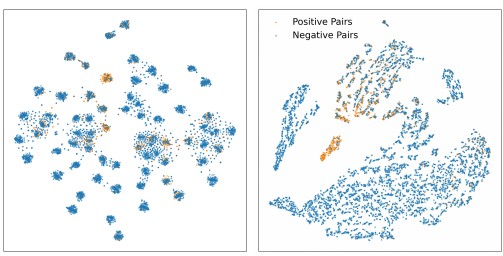

Figure 19: t-SNE visualization for TF MED12. Left: Input embeddings based on raw expression. Right: Model-learned output embeddings. Target genes form tighter clusters post-training.

Figure 20: Global t-SNE visualization of all TF–target pairs for the Specific hESC dataset. Left: input space from raw expression. Right: output space from SCUNIGP. Clear class-wise separation is observed.

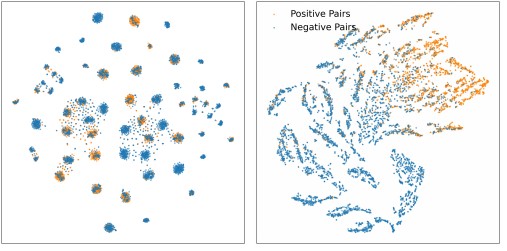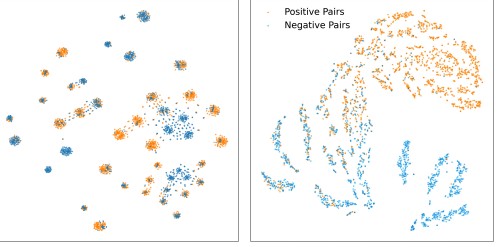

Figure 21: Global t-SNE visualization of all TF–target pairs for the Specific hHEP dataset. Left: input space from raw expression. Right: output space from SCUNIGP. Clear class-wise separation is observed.

Figure 22: Global t-SNE visualization of all TF–target pairs for the Specific mHSC-GM dataset. Left: input space from raw expression. Right: output space from SCUNIGP. Clear class-wise separation is observed.

