# OpenReview forum: "Unifying Graph-Based and Pairwise-Based Representations for Gene Regulatory Network Inference from scRNA-seq Data"
_ICLR.cc/2026/Conference — ICLR 2026 Conference Withdrawn Submission_

### Official Review · Reviewer_Nodb · 2025-10-24

**Soundness:** 2
**Presentation:** 3
**Contribution:** 2
**Rating:** 6
**Confidence:** 5

**Summary:**

This paper introduces scUniGP, a novel and compelling framework for Gene Regulatory Network (GRN) inference from scRNA-seq data. The work directly addresses the fundamental, complementary limitations of existing GRN inference approaches: the overfitting of graph-based models dueictated to reliance on a single training graph, and the failure of pairwise-based methods to capture global topological structure. The central innovation is the unification of these two paradigms through a two-stage, multi-scale fusion architecture.

**Strengths:**

1. Unified and Robust Architecture: scUniGP's strength lies in its ability to simultaneously model global and local regulatory signals.
2. Improve the SOTA performance
3. Interpretability and representational learning

**Weaknesses:**

1. Scale and Scope of Datasets (e.g., whole genome scale GRN)
2. Model architecture and hyperparameter setting
3. Rigor of the benchmarking strategy

**Questions:**

1. Traditional methods also include Matrix Factorization (MF). Please include a discussion and citation for Matrix Factorization methods in the Related Work section (e.g., cite: https://doi.org/10.1073/pnas.2136632100; https://www.genome.org/cgi/doi/10.1101/gr.265595.120; https://doi.org/10.1038/s41540-024-00386-w).
2. Could the authors clarify how the hyperparameters were chosen and determined?Benchmarking protocol lacks consistency.
3. Why were only the datasets from the BEELINE framework used, and not the complete BEELINE benchmarking protocol?
4. Can the model be scaled to the whole-genome level ($\sim 1,600$ TFs and $\sim 20,000$ genes)? What would be the associated computation time and accuracy at this scale?
5. In addition to the hard negative sampling results, could the authors provide results for naive negative sampling or the results without any sampling (i.e., using the full sparse pool), perhaps in the Appendix?
6. Since some models utilize a prior network while others do not, is the current comparative evaluation fair? Could the authors clearly label this information in the comparison table?
7. In practical application, which specific prior network should be used? Given that all prior networks are inherently biased by current human knowledge (potentially containing many false negatives), what is the impact of this bias on the model's final results?

---

### Official Review · Reviewer_4nrn · 2025-10-30

**Soundness:** 3
**Presentation:** 3
**Contribution:** 2
**Rating:** 4
**Confidence:** 3

**Summary:**

This manuscript addresses the task of gene regulatory network (GRN) inference. The authors aim to overcome limitations of both graph-based and pairwise-based approaches by proposing a unified framework, scUniGP, which jointly models global regulatory topology and local TF–target interactions. Extensive experiments on seven benchmark datasets demonstrate the superior performance of the proposed model compared to existing methods.

**Strengths:**

This paper is very well-written and well-organized, presenting complex concepts clearly and logically, which makes the methodology easy to follow. The experimental setup is comprehensive and thoughtfully designed, evaluating the model on seven benchmark GRN datasets and comparing against state-of-the-art graph- and pairwise-based methods, providing convincing evidence of the model’s effectiveness.

**Weaknesses:**

The first concern is that the novelty of the proposed framework may be limited for a venue like ICLR, although the motivation, integrating global topological modeling with local TF–target interactions, is quite interesting and potentially impactful if this integration is novel. The second concern is the relatively small size of the benchmark datasets used in the experiments, with the number of genes ranging from 560 to 1120 and cells from 383 to 1071, which raises questions about the scalability and practical applicability of the proposed model to larger, more complex datasets.

**Questions:**

Please see the weaknesses section for further discussion (technological novelty, scalability and practical applicability).

---

### Official Review · Reviewer_j2yG · 2025-11-01

**Soundness:** 2
**Presentation:** 2
**Contribution:** 1
**Rating:** 2
**Confidence:** 4

**Summary:**

This paper proposes a method that integrates graph-based and pairwise-based representations for GRN inference.

**Strengths:**

1. integration of graph-based and pairwise-based representations

**Weaknesses:**

1. Experimental details are missing. Which prior GRN is used?
2. The comparison results are unfair because many of the competing methods do not have prior GRN. It is hard to know the outperformance is coming from the prior or the method itself.

**Questions:**

Check weakness.

---

### Official Review · Reviewer_ujF6 · 2025-11-03

**Soundness:** 3
**Presentation:** 3
**Contribution:** 2
**Rating:** 4
**Confidence:** 4

**Summary:**

The paper is focused on gene regulatory network inference problem, that is, inferring edges in a graph with genes (or transcription factors they encode) as nodes. The authors propose a novel method that integrates global, GNN approach and local, edge-by-edge modeling.

**Strengths:**

The proposed method is an original, if somewhat incremental: it takes two well-established approaches, and adds a fusion component that unifies the results from them. The Global Branch utilizes GCN or GAT graph models to generate graph-level embeddings based on expression data for each node. Separately the edge-centric, TF-to-target-gene Pairwise Branch generates gene expression-based embeddings using a small Transformer encoder model. The final fusion step uses a multi-layer model that operates on edge level, takes input from the pairwise step for that edge, as well as graph-level embeddings for the nodes connected by the edge from the corresponding layer in the GNN; in the last layer, a global pairwise score from the GNN is also used. Training of the pipeline proceeds in two phases: first the GNN is trained, then the full model, including pairwise branch and fusion model, as well as GNN, are fine-tuned.

The model is verified experimentally on a broad set of benchmark datasets, and compared with nine competing methods. It achieves higher scores than all the competing methods.

The paper is clearly written and easy to follow.

**Weaknesses:**

The originality of the fusion approach is somewhat limited, and similar approaches have been proposed recently, e.g. scRegNet [1], which uses a fusion of Transformer and GNN, and achieves similar or in some cases higher performance scores (although the exact test methodology may differ somewhat).

[1] Kommu et al., Prediction of Gene Regulatory Connections with Joint Single-Cell Foundation Models and Graph-Based Learning, bioRxiv: 2025 Jan 29:2024.12.16.628715, Proc. Of ISMB/ECCB 2025.


The ablation study is not fully complete, it does indicate that pairwise method benefits from fusion with GNN, but the opposite question, does GNN benefit from fusion with pairwise information ($z_{ij}^{(0)}$ in eq (7)) is not explored; what would happen if it was set to $z_{ij}^{(0)}=0$, or to a non-pairwise-branch, GNN-only-derived term similar to eq (5): $z_{ij}^{(0)} = {h_i^{(1)}}^T h_j^{(1)}$. That is, if a separate pairwise model (eq 6) was not used at all, only the pretrained global branch (eq 4-5) and fine-tuned multi-layer fusion (eq 7-8)?

While the paper is well written in general, there are some places that could be phrased better. For example, in outlining the weaknesses of pairwise methods, the authors write “
if transcription factor g1→g2, and g2→g3, the indirect regulatory influence
of g1 on g3 can be overlooked by pairwise-based methods”. It seems to imply that the goal of the GRN is to place a single edge over multi-hop interactions, but actually the challenge for pairwise methods is that they will detect the influence and incorrectly predict direct g1→g3 edge, because they lack the global view to recognize the actual g1→g2→g3 indirect path.

**Questions:**

How does the method compare to scRegNet?

How important is the fusion with the pairwise branch? What would the results be with $z_{ij}^{(0)}=0$ and with $z_{ij}^{(1)} = {h_i^{(1)}}^T h_j^{(1)}$

---

### Note · Authors · 2026-01-06

I have read and agree with the venue's withdrawal policy on behalf of myself and my co-authors.